# The Promise of Semantic Segmentation in Detecting Actinic Keratosis Using Clinical Photography in the Wild

**DOI:** 10.3390/cancers15194861

**Published:** 2023-10-05

**Authors:** Panagiotis Derekas, Panagiota Spyridonos, Aristidis Likas, Athanasia Zampeta, Georgios Gaitanis, Ioannis Bassukas

**Affiliations:** 1Department of Computer Science & Engineering, School of Engineering, University of Ioannina, 45110 Ioannina, Greece; panderekas@gmail.com (P.D.); arly@cs.uoi.gr (A.L.); 2Department of Medical Physics, Faculty of Medicine, School of Health Sciences, University of Ioannina, 45110 Ioannina, Greece; 3Department of Skin and Venereal Diseases, Faculty of Medicine, School of Health Sciences, University of Ioannina, 45110 Ioannina, Greece; athanasiazampeta@gmail.com (A.Z.); ggaitan@uoi.gr (G.G.); ibassuka@uoi.gr (I.B.)

**Keywords:** deep learning, semantic segmentation, U-Net, actinic keratosis, cutaneous cancerization field, skin lesions, clinical photography

## Abstract

**Simple Summary:**

Understanding the relationship between the skin cancerization field and actinic keratosis (AK) is crucial for identifying high-risk individuals, implementing early interventions, and preventing the progression to more aggressive forms of skin cancer. Currently, the clinical tools for grading field cancerization primarily involve assessing AK burden. In addition to their inherent subjectivity, these grading systems are limited by the high degree of AK lesions’ recurrence. The present study proposes a method based on deep learning and semantic segmentation to improve the monitoring of the AK burden in clinical settings with enhanced automation and precision. The experimental results highlight the effectiveness of the proposed method, paving the way for more effective and reliable evaluation, continuous monitoring of condition progression, and assessment of treatment responses.

**Abstract:**

AK is a common precancerous skin condition that requires effective detection and treatment monitoring. To improve the monitoring of the AK burden in clinical settings with enhanced automation and precision, the present study evaluates the application of semantic segmentation based on the U-Net architecture (i.e., AKU-Net). AKU-Net employs transfer learning to compensate for the relatively small dataset of annotated images and integrates a recurrent process based on convLSTM to exploit contextual information and address the challenges related to the low contrast and ambiguous boundaries of AK-affected skin regions. We used an annotated dataset of 569 clinical photographs from 115 patients with actinic keratosis to train and evaluate the model. From each photograph, patches of 512 × 512 pixels were extracted using translation lesion boxes that encompassed lesions in different positions and captured different contexts of perilesional skin. In total, 16,488 translation-augmented crops were used for training the model, and 403 lesion center crops were used for testing. To demonstrate the improvements in AK detection, AKU-Net was compared with plain U-Net and U-Net++ architectures. The experimental results highlighted the effectiveness of AKU-Net, improving upon both automation and precision over existing approaches, paving the way for more effective and reliable evaluation of actinic keratosis in clinical settings.

## 1. Introduction

A cutaneous or skin cancerization field (SCF) refers to the medical condition wherein the chronic ultraviolet radiation (UVR)-damaged skin area around incident tumors or precancerous lesions harbors clonally expanding cell subpopulations with distinct pro-cancerous genetic alterations [1]. As a result, this skin area becomes more susceptible to the development of new similar malignant lesions or the recurrence of existing ones [2]. The identification of an SCF mainly relies on recognizing actinic keratosis (AK) [3]. AK, also known as keratosis solaris, is a common skin condition caused by long-term sun exposure. These pre-cancerous lesions not only serve as visible markers of chronic solar skin damage but, if left untreated, each of them can possibly progress into a potentially fatal cutaneous squamous cell carcinoma (CSCC) [4,5]. Since there are no established prognostic factors to predict which individual AK will progress into CSCC, early recognition, treatment, and follow-up are vital and endorsed by international guidelines to minimize the risk of invasive CSCC [5,6,7]. On the other hand, AKs, more than being the hallmark lesions of UVR-damaged skin at risk to develop skin cancers, because of their visual similarity to invasive neoplasia, also constitute a per se important macromorphological differential diagnostic challenge in a skin cancer screening setting. In the everyday clinical setting, dermatologists employ a spectrum of established non-invasive diagnostic techniques to increase the sensitivity and specificity of the clinical diagnostic workout and to improve the discrimination between AK and CSCC [3,8]. Besides the widely available and routinely applied dermoscopy to discriminate between invasive CSSC and AK [9], recent findings show that the combination of minimally invasive histologic markers, like the basal layer proliferation score, [10] with non-invasive imaging modalities, like LC-OCT, can be used to better stratify AKs according to their risk to progress to CSCC [11].

Understanding the relationship between field cancerization and AK is crucial for identifying high-risk individuals, implementing early interventions, and preventing the progression to more aggressive forms of skin cancer [12]. Currently, the clinical tools for grading SCF primarily involve assessing AK burden using systems like AKASI [13] and AK-FAS [14]. Notably, the latter approach is based on the evaluation of standardized clinical photographs of preselected, target skin areas to reduce the considerable inter-observer subjectivity of the clinical examination. However, it is still based on subjective evaluation by trained physicians who localize, assess, and count “visible” AK lesions on the available photographs [14]. Particularly, the high inter-observer variability in counting lesions seriously limits the reliability of the above approaches [15,16]. In addition to their inherent subjectivity, these grading systems are limited by their high degree of spatial plasticity, including the trend to recurrence of individual AK lesions. They are actually snapshots of the AK burden of a certain skin site, unable to track whether the recorded lesions are relapses of incident AK at the same site or newly emerging lesions arising de novo in nearby areas [17].

Clinical photography is a low-cost, portable solution that efficiently stores “scanned” visual information of an entire skin region. Considering the pathobiology of AK, Criscione et al. applied an image analysis-based methodology to analyze clinical images of chronically UV-damaged skin areas in a milestone study [18]. With this approach, they could estimate the risk of progression of AKs to keratinocyte skin carcinomas (KSCs) and assess the natural history of AK evolution dynamics for approximately six years. Since then, the evolving efficiency of deep learning algorithms in image recognition and the availability of extensive image archives have greatly accelerated the development of advanced computer-aided systems for skin disease diagnosis [19,20,21,22]. Particularly for AK, given the multifaced role of AK recognition in cutaneous oncology, the assignment of a distinct diagnostic label to this condition was also a core task in many studies that applied machine learning as a tool to differentiate skin diseases based on clinical images of isolated skin lesions [23,24,25,26,27,28,29,30,31]. 

However, despite significant progress in image analysis of cropped skin lesions in recent years, the determination of the AK burden in larger, sun-affected skin areas still remains a quite challenging diagnostic task. AKs present as pink, red, or brownish patches, papules, or flat plaques or may even have the same color as the surrounding skin. They vary in size from a few millimeters to 1–2 cm and can either be isolated or more typically, numerous, sometimes even widely confluent in some patients. Moreover, the surrounding skin may show signs of chronic sun damage, including telangiectasias, dyschromia, and elastosis [6] and also seborrheic keratoses and related lesions, notably lentigo solaris. These latter lesions are the most frequently encountered benign growths within an SCF and underlie most of the confusion related to the clinical differential diagnosis of AK lesions [32]. All these visual features add substantial complexity to the task of automated AK burden determination.

Earlier approaches to automated AK recognition from clinical images were limited either by treating AK as a skin erythema detection problem [33] or by restricting the detection to preselected smaller sub-regions of wider photographed skin areas, using a binary patch classifier for AK discrimination from healthy skin [34]. Moreover, these approaches are prone to a large risk of false-positive detections since erythema is present in various unrelated skin conditions, and the contamination of the images by diverse concurrent and confluent benign growths [35,36] seriously affects the accuracy of binary classification schemes to estimate AK burden in extended skin areas.

To tackle the demanding task of AK burden detection in UVR-chronically exposed skin areas, more recent work has proposed a superpixel-based, convolutional neural network (CNN) for AK detection (i.e., AKCNN) [37]. The core engine of AKCNN is a patch classifier, a lightweight CNN, adequately trained to distinguish AK from healthy skin but also from seborrheic keratosis and solar lentigo. However, a main limitation of AKCNN that persists is the manual preselection of the area to scan, excluding image parts with “disturbing” visual features, like the hairs, nose, lips, eyes and ears, which are found to result in false-positive detections.

To improve the monitoring of AK burden in clinical settings with enhanced automation and precision, the present study evaluates the application of semantic segmentation using the U-Net architecture [38], with transfer learning to compensate for the relatively small dataset of annotated images, and a recurrent process to efficiently exploit contextual information and address the specific challenges related to the low contrast and ambiguous boundaries of AK-affected skin regions.

Although there has been considerable research on deep learning-based skin lesion segmentation, the primary focus has been delineating melanoma lesions in dermoscopic images [39,40]. The present study is a further development of our previous research on AK detection and, to the best of our knowledge, is the first study to employ semantic segmentation to clinical images for AK detection and SCF evaluation. Our main aim herewith is to contribute to the development of reliable instruments to assess AK, and, consequently, SCF burden, to be primarily used in the evaluation of therapeutic interventions.

## 2. Materials and Methods

### 2.1. Methods

#### 2.1.1. Overview of Semantic Segmentation and U-Net Architecture

Today, with the advent of deep learning and convolutional neural networks, semantic segmentation goes beyond traditional segmentation by associating a meaningful label to each pixel in an image [41]. Semantic segmentation plays a crucial role in medical imaging, and it has numerous applications, including image analysis and computer-assisted diagnosis, monitoring the evolution of conditions over time, and assessing treatment responses.

A groundbreaking application in the field of semantic segmentation is the U-Net architecture, which emerged as a specialized variant of the fully convolutional network architecture. This network architecture excels in capturing fine-grained details from images and demonstrates effectiveness even with limited training data, making it well suited for the challenges posed by the field of biomedical research [38]. The U-Net architecture is characterized by its “U” shape, with symmetric encoder and decoder sections (Figure 1).

The encoder section consists of a series of convolutional layers with 3 × 3 filters, followed by rectified linear unit (ReLU) activations. After each convolutional layer, a 2 × 2 max pooling operation is applied, reducing its spatial dimensions while increasing its number of feature channels. As the encoder progresses, the spatial resolution decreases while the number of feature channels increases. This allows the model to capture increasingly abstract and high-level features from the input image.

The decoder section of the U-Net starts with an upsampling operation to increase the spatial dimensions of the feature maps. The upsampling is typically performed using transposed convolutions. At each decoder step, skip connections are introduced. These connections directly connect the feature maps from the corresponding encoder step to the decoder, preserving fine-grained details and spatial information. The skip connections are achieved by concatenating the feature maps from the encoder with the upsampled feature maps in the decoder.

In the following, the concatenation, the combined feature maps go through convolutional layers with 3 × 3 filters and ReLU activations to refine the segmentation predictions. Each decoder step typically reduces the number of feature channels to match the desired output shape.

Finally, a 1 × 1 convolutional layer is applied to produce the final segmentation map, with each pixel representing the predicted class label.

Since its introduction, U-Net has served as a foundation for numerous advancements in medical image analysis and has inspired the development of various modified architectures and variants [42,43].

Transfer learning has significantly contributed to the success of U-Net in various medical image segmentation tasks [44,45,46,47]. Initializing the encoder part of U-Net with weights learned from a pretrained model enables U-Net to start with a strong foundation of learned representations, accelerating convergence, and improving segmentation performance, especially when the availability of annotated medical data is limited.

In this study, we used the VGG16 model [48], pretrained on ImageNet [49], as the backbone for the encoder in the U-Net architecture. Both U-Net and VGG16 are based on the concepts of deep convolutional neural networks, which are designed to encode hierarchical representations from images enabling them to capture increasingly complex features. Both architectures utilize convolutional layers with 3 × 3 filters followed by rectified linear unit activations and a 2 × 2 max pooling operation for downsampling as their primary building blocks.

Figure 2 illustrates our transfer learning scheme, where four convolution blocks of pretrained VGG16 were used for the U-Net encoder.

#### 2.1.2. Batch Normalization

Batch normalization (BN), proposed by Sergey Ioffe and Christian Szegedy [50], is a technique commonly used in neural networks to normalize the activations of a layer by adjusting and scaling them. It helps to stabilize and speed up the training process by reducing the internal covariate shift, which refers to the change in the distribution of the layer’s inputs during training. The BN operation is typically performed by computing the mean and variance of the activations within a mini-batch during training. These statistics are then used to normalize the activations by subtracting the mean and dividing by the square root of the variance. Additionally, batch normalization introduces learnable parameters, known as scale and shift parameters, which allow the network to adaptively scale and shift the normalized activations [50].

Since each mini-batch has a different mean and variance, this introduces some random variation or noise to the activations. As a result, the model becomes more robust to specific patterns or instances present in individual mini-batches and learns to generalize better to unseen examples.

In U-Net, the decoding layers are responsible for reconstructing the output image or segmentation map from the upsampled feature maps. By applying BN to the decoding layers in our U-Net, the regularization effect is targeted to the reconstruction process, improving the model’s generalization ability and reducing the risk of overfitting in the output reconstruction. 

#### 2.1.3. ConvLSTM: Spatial Recurrent Module in U-Net Architecture

The recurrent mechanisms in neural network architecture (i.e., RNN) have been adapted to work with sequential data, with decisions based on current and previous inputs. There are different kinds of recurrent units based on how the current and last inputs are combined, such as gated recurrent units and long short-term memory (LSTM) [51]. The LSTM module was designed by introducing three gating mechanisms that control the flow of information through the network: the input gate, the forget gate, and the output gate. These gates allow the LSTM network to selectively remember or forget information from the input sequence, which makes it more effective for long-term dependencies. For text, speech, and signal processing, plain RNNs are directly used, while for use in 2D or 3D data (e.g., images), RNNs have been extended correspondingly using convolutional structures. Convolutional LSTM (convLSTM) is the LSTM counterpart for long-term spatiotemporal predictions [52].

In semantic segmentation, recurrent modules have been incorporated in various ways in U-Net architecture. Alom et al. [53] introduced recurrent convolutional blocks at the backbone of the U-Net architecture to enhance the ability of the model to integrate contextual information and improve feature representations for medical image segmentation tasks. In recent work, Arbelle et al. [54] proposed the integration of convLSTM blocks at every scale of the encoder section of the U-Net architecture, enabling multi-scale spatiotemporal feature extraction and facilitating cell instant segmentation in time-lapse microscopy image sequences. 

In the U-Net architecture, max pooling is commonly used in the encoding path to downsample the feature maps and capture high-level semantic information. However, max pooling can result in a loss of spatial information and details between neighboring pixels. Several researchers in the field of skin lesion segmentation using dermoscopic images have exploited recurrent layers as a mechanism to refine the skip connection process of U-Net [55,56,57]. In the present study, to address the specific challenges related to the low contrast and ambiguous boundaries of AK-affected skin regions, we employed convLSTM layers to bridge the semantic gap between the feature map extracted from the encoding path $X_{e}^{l}$ and the output feature map after upsampling in the decoding path $X_{d}^{l,up}$. We assume $X_{l}\in R^{F_{l}\;x{\;h}_{l}\;x\;w_{l}}$ is the concatenation of $X_{e}^{l}$ and $X_{d}^{l,up}$ where $F_{l}$ is the number of filters and $h_{l},\;w_{l}$ are the height and width of the feature map. $X_{l}$ is split into n x m patches $P_{i,j}\in R^{F_{l}\;x{\;h}_{p}\;x\;w_{p}}$ where
(1)$$h_{p}=h_{l}/n$$and (2)$$\;w_{p}=w_{l}/m$$

Following this, we will refer to $P_{i,j}$ as $P_{t}$ and replace subscripts *i*,*j* with the notion of processing step *t*. The input $P_{t}$ is passed through the convolutional operation to compute three gates that regulate the spatial information flow: the input gate $i_{t}$, the forget gate $f_{t}$, and the output gate $o_{t}$ as follows.
(3)$$i_{t}=\sigma \left(W_{xi}*P_{t}+W_{hi}*H_{t-1}+W_{ci}\cdot c_{t-1}+b_{i}\right)$$
(4)$$f_{t}=\sigma \left(W_{xf}*P_{t}+W_{hf}*H_{t-1}+W_{cf}\cdot c_{t-1}+b_{f}\right)$$
(5)$$o_{t}=\sigma \left(W_{xo}*P_{t}+W_{ho}*H_{t-1}+W_{co}\cdot c_{t}+b_{o}\right)$$

ConvLSTM also maintains the cell state ($c_{t}$) and the hidden state ($H_{t}$):(6)$$c_{t}=f_{t}\cdot c_{t-1}+i_{t}\cdot \tanh\left(W_{xc}*P_{t}+W_{hc}*H_{t-1}+b_{c}\right)$$
(7)$$H_{t}=o_{t}\cdot tanh(c_{t})$$
where $W_{x*}$ and $W_{h*}$ correspond to the 2D convolutional kernel of the input and hidden states, respectively. * represents the convolutional operation and bullet (·), the Hadamard function (element-wise multiplication), respectively. $b_{i},b_{f},b_{c}$, and $b_{o}$ are the bias terms and sigma is the sigmoid function. 

Figure 3 depicts the model for AK detection based on U-Net architecture (AKU-Net) that incorporates transfer learning in the encoding path, the recurrent process in skip connections, and the BN in the decoding path.

#### 2.1.4. Loss Function

In this study, we approached AK detection as a binary semantic segmentation task, where each pixel in the input image was classified as either being in the foreground (belonging to the skin areas affected by AK) or background. For this purpose, we utilized the binary cross-entropy loss function, commonly used in U-Net, also known as the log loss or sigmoid loss.

Mathematically, the binary cross-entropy loss for a single pixel can be defined as the following:(8)L(p, y)=−y ∗ log(p)−(1−y) ∗ log(1−p)
where p is the predicted probability of the pixel belonging to the foreground class, obtained by applying a sigmoid activation function to the model’s output, and y is the ground truth label (0 for background, 1 for foreground) indicating the true class of the pixel.

The overall loss for the entire image was then computed as the average of the individual pixel losses. During training, the network aimed to minimize this loss function by adjusting the model’s parameters through backpropagation and gradient descent optimization algorithms.

#### 2.1.5. Evaluation

To demonstrate the expected improvements in AK detection, AKU-Net was compared with plain U-Net and U-Net^++^ [58]. U-Net^++^ comprises an encoder and decoder connected through a series of nested dense convolutional blocks. The main concept behind U-Net^++^ is to narrow the semantic gap between the encoder and decoder feature maps before fusion. The authors of U-Net^++^ reported superior performance compared to the original U-Net in various medical image segmentation tasks, including electron microscopy cells, nuclei, brain tumors, liver, and lung nodules.

To ensure comparable results, the trained networks utilized the VGG16 as the backbone and a BN in the decoding path. The segmentation models were assessed by means of the Dice coefficient and Intersection over Union (IoU).

The Dice coefficient measures the similarity or overlap between the predicted segmentation mask and the ground truth mask. The formula for the Dice coefficient is the following:(9)$$Dice=\frac{\left(2*\left|A\;\cap \;B\right|\right)}{\left(\left|A\right|+\left|B\right|\right)}=\frac{2*TP}{2*TP+\;FN\;+FP}$$

The IoU measures the intersection between the predicted and ground truth masks relative to their union. The formula for the IoU is the following:(10)$$IoU=\frac{\left|A\;\cap \;B\right|}{\left|A\;\cup \;B\right|}=\frac{TP}{TP+FN+FP}$$

The notion | | represents the total number of pixels. TP, FN, FP are the true-positive, false-negative, and false-positive prediction rates at the pixel level, respectively.

To provide comparison results with a recent work on AK detection, we also employed the adapted region-based F1 score $\left(aF1\right)$ [37]:(11)$$aF1=\frac{2*aRec*\;aPrec}{aRec+aPrec}$$

The aF1 score was introduced by the authors to compensate for the fact that AK lesions often lack sharply demarcated borders, and experienced clinicians can provide only rough, approximate AK annotations in clinical images. The adapted estimators of Recall (aRec) and Precision (aPrec) are estimated as follows.

Assuming a ground truth set of N annotated (labeled) areas, ${AK}_{area}=\left\{{AK}_{1,}{AK}_{2},\ldots {AK}_{N}\right\}$, and the set of pixels predicted as AK by the system, AKpred, we define
(12)$${TPC}_{i}=\left\{\begin{array}{c} 0\;if\;{AK}_{i}\cap \;AKpred=\emptyset \\ 1\;if\;\;{AK}_{i}\cap \;AKpred\neq \emptyset \end{array}\right.$$
and the True Positive Counts (TPC):(13)$$TPC=\sum_{i=1}^{N}{TPC}_{i}\;\;$$

The adapted estimators aRec and aPrec are given as the following:(14)$$aRec=\frac{true\;positive\;counts}{actual\;positive\;counts}=\frac{TPC}{N}\;$$
(15)$$aPrec=\frac{True\;positive\;area}{Total\;predicted\;positive\;area}=\frac{\left|{AK}_{area}\cap AKpred\right|}{\left|{AKCNN}_{area}\right|}$$

All coefficients, the Dice, the IoU, and the aF1, range from 0 to 1. Higher values of these metrics indicate greater similarity between the predicted and ground truth masks and, consequently, better performance.

Figure 4 provides a qualitative example of the error tolerance in estimating aF1, favoring compensation for rough annotations.

#### 2.1.6. Implementation Details

The segmentation models were implemented in Python 3, adopting an open-source framework in Keras API with TensorFlow as the backend. The experimental environment was based on a Windows 10 workstation configured with an AMD 9 series 3900X CPU@3.80 GHz processor, 64 GB of 3200 MHz DDR4 ECC RDIMM, NVIDIA RTX 3070 GPU memory of 8 GB. The Adam optimizer $\left(learningrate=0.001,weightdecay=10^{-6}\right)$ was employed to train the networks. Considering the constraints of our computational environment, we set the batch size equal to 32 and the number of iterations to 100.

### 2.2. Materials 

The use of archival photographic materials for this study was approved by the Human Investigation Committee (IRB) of the University Hospital of Ioannina (Approval No.: 3/17-2-2015(θ.17)). The study included a total of 115 patients (60 males, 55 females; age range: 45-85 years) with facial AK who attended the specialized Dermato-oncology Clinic of the Dermatology Department. 

Facial photographs were acquired with the camera axis perpendicular to the photographed regions. The distance was adjusted to include the whole face from the chin to the top of the hair/scalp border. 

Digital photographs of a 4016 × 6016 pixel spatial resolution were acquired according to a procedure adapted from Muccini et al. [43], using a Nikon D610 (Nikon, Tokyo, Japan) camera with a Nikon NIKKOR^©^ 60 mm 1:2.8G ED micro lens mounted on it. The camera controls were set at f18, with a shutter speed of 1/80 s, ISO 400, autofocus, and white balance at auto-adjustment mode. A Sigma ring flash (Sigma, Fukushima, Japan) at TTL mode was mounted to the camera. In front of the lens and the flashlight, linear polarized filters were appropriately adapted to ensure a 90° rotation of the ring flash polarization axis to the relevance of the lens-mounted polarizing filter. For the AK annotation, two physicians (GG and AZ) jointly discussed and reached an agreement on the affected skin regions. Notably, cross-polarized photography was employed that enhanced the visibility of the vascular plexus of the skin (redness) and removed unwanted glare from the epidermis, allowing for the detailed evaluation of the assessed area at larger magnifications, as required. For the purposes of the present study, figures were eligible only from patients without a history of CSSC and from areas without ambiguous lesions, including pigmented lesions, clinically susceptive to cutaneous malignancy. Assessments of the clinical grades of individual lesions are not reported, as they are out of the scope of the present study. However, we incorporated lesions independent of clinical AK severity grading [59] such that the lesions could be suspected on the selected clinical photographs, as well as lesions with variable degrees of pigmentation. This also applied to AKs in extra-facial anatomical regions.

Multiple photographs were taken per patient to capture the presence of lesions across the entire face. Additionally, these photographs were intended to provide different views of the same lesions, resulting in 569 annotated clinical photographs (Figure 5).

Patches of 512 × 512 pixels were extracted from each photograph using translation lesion boxes. These boxes encompassed lesions in various positions and captured different contexts of the perilesional skin. In total, 16,891 lesion center and translation-augmented patches were extracted (Figure 6).

#### Experimental Settings

Since multiple samples (skin patches) from the same patient were used to prevent data leakage in train validation test datasets and ensure unbiased evaluations of the models’ generalization, we performed dataset splitting at the patient level. We used 510 photographs obtained from 98 patients for training, extracting 16,488 translation-augmented image patches. Among them, approximately 20% of the crops (3298 patches) from 5 patients were reserved for validation. An independent set consisting of 17 patients (59 photographs) was used to assess the model’s performance, yielding 403 central lesion patches (Table 1).

It is important to note that since the available images had a high spatial resolution, we had to train the model using rectangular crops. The size of these crops was selected to include sufficient contextual information. To achieve this, the images were first rescaled by 0.5, and crops of a size 512 × 512 were extracted. However, due to computational limitations, the patches were further rescaled by a factor of 0.5 (256 × 256 pixels). Using an internal fiducial marker [13], we estimated that our model was ultimately trained with cropped images at a scale of approximately ~5 pixels/mm.

## 3. Results

Dice and IoU coefficients were utilized to evaluate the segmentation accuracy of the model. Table 2 summarizes the comparison results employing the standard U-Net and U-Net^++^ architectures.

The AKU-Net model demonstrated a statistically significant improvement in AK detection compared to U-Net^++^ (*p* < 0.05; Wilcoxon signed-rank test). It is worth noting that the standard U-Net model had limitations in detecting AK areas, as it could not detect AK in 257 out of the total 403 testing crops, corresponding to approximately 63% of the testing cases. An exemplary qualitative comparison of the goodness of the segmentation of incident AK with the different model architectures is illustrated in Figure 7.

In a recent study, we implemented the AKCNN model for efficient AK detection [37] in manually preselected wide skin areas. To evaluate the performance of the AKU-Net model in broad skin areas and compare it with the AKCNN, we used the same evaluation set as the one employed in the previous study [37]. 

The AKU-Net model was trained using patches of 256 × 256 pixels. However, to allow the present model to be evaluated and compared with the AKCNN, we performed the following steps:Zero Padding: Appropriate zero-padding was applied to the input image to make it larger and suitable for subsequent cropping.Image Cropping: The zero-padded image was divided into crops of 256 × 256 pixels. These crops were used as individual inputs to the AKU-Net model for AK detection.Aggregating Results: The obtained segmentation results for each 256 × 256-pixel crop were combined to obtain the overall AK detection for the entire broad skin area.

Table 3 compares the performances (accuracy measures) of the AKU-Net and AKCNN model architectures (for *n* = 10 random frames). At a similar image scale of approximately ~7 pixels/mm, there were no significant differences in the levels of model performance in terms of aPrec (*p* = 0.6), aRec (p=0.4), and aF1 (p=0.6; Wilcoxon signed-rank test).

Although AKU-Net exhibited AK detection accuracy at the same level as that of AKCNN, the substantial advantage of AKU-Net is that the latter does not require the manual preselection of scanning areas. Explanative examples are given in Figure 8 and Figure 9.

## 4. Discussion

In this study, we utilized deep learning, specifically in the domain of semantic segmentation, to enhance the detection of actinic keratosis (AK) on clinical photographs. In our previous approach, we introduced a CNN patch classifier (i.e., AKCNN) to assess the burden of AK in large skin areas [51]. Detecting AK in regions with field cancerization presents challenges, and a binary patch (regional) classifier, specifically “AK versus all”, had serious limitations. AKCNN was implemented to effectively distinguish AK from healthy skin and differentiate it from seborrheic keratosis and solar lentigo, thereby reducing false-positive detections. However, AKCNN is subject to manually predefined scanning areas which are necessary to exclude skin regions prone to false diagnosis as AK on clinical images (Figure 8 and Figure 9).

To enhance the monitoring of the AK burden in real clinical settings with improved automation and precision, we utilized a semantic segmentation approach based on an adequately adapted U-Net architecture. Despite using a relatively small dataset of weakly annotated clinical images, the AKU-Net exhibited a remarkably improved performance, particularly in challenging skin areas. The AKU-Net model outperformed the corresponding baseline models, the U-Net and U-Net++, highlighting the efficiency boost achieved through the spatial recurrent layers added in skip connections.

Considering its scanning performance, AKU-Net was evaluated by aggregating its scanning results from 256 × 256-pixel crops and comparing them with those from the AKCNN (Table 3). Both approaches exhibited a similar level of recall (true-positive detection). However, AKU-Net is favorably tolerant of the selection of the scanning area, which can simply be a boxed area that includes the target region, providing false-positive rates at least comparable to those obtained with the AKCNN on manually predefined scanning regions. 

It is important to note that due to computational constraints, the AKU-Net was trained on image crops of 256 × 256 pixels, at a scale of about ~5 pixels/mm. For this, the original image was subsampled twice. This subsampling process resulted in a degradation of spatial resolution, which imposed limitations on the system’s ability to detect AK lesions (recall level). This limitation is also supported by evidence from our previous studies: in a similar scale of ~7 pixels/mm, AKCNN experienced a drop in recall [37]. We expect a significantly improved AK detection accuracy by subsampling the original image and training the network with crops of 512 × 512 pixels.

The palpation of the lesional skin to confirm the characteristic “sandpaper” sign is crucial for the diagnosis of barely visible, flat, grade I AKs. However, the fact that the present approach relies on photographic materials that might theoretically lead to underestimation of the AK burden does not represent a serious limitation of this method. It is worth noting that the “sandpaper” sign has been ranked as a less reliable feature of the sun damage of SCF skin areas compared to “visible” features (telangiectasia, atrophy, and pigmentation disorders) in a panel of experts’ study [3]. Moreover, the proposed approach, like AK-FAS too [14], primarily aims to quantify the AK burden in selected skin areas to assist with the evaluation of therapeutic interventions. Accordingly, the burden measurements are based on the evaluation of index lesions in the preselected target area. If required, the latter can be planned to include “hidden” areas, like the retroauricular area or skin regions covered by hair. 

Future efforts will explore the optimal trade-off between image scale and the size of the input crop used to train the network, which could lead to superior detection of AK lesions. Moreover, implementing a system with known restrictions, that is, knowing the range of acceptable image scales, is essential for future studies to complement and validate the system’s generalizability utilizing multicenter datasets from various cameras.

In future studies, it is also essential to evaluate the system’s accuracy in relation to the variability among experts when recognizing AK using clinical photographs. This assessment is crucial as it will gauge the system’s performance compared to the varying detections of human experts.

## 5. Conclusions

Understanding the relationship between the biology of a skin cancerization field and the burden of incident AK is crucial to identifying high-risk individuals, implementing early interventions, and preventing the development of more aggressive forms of skin cancer. 

The present study evaluated the application of semantic segmentation based on the U-Net architecture to improve the monitoring of the AK burden in clinical settings with enhanced automation and precision. Deep learning algorithms for semantic image segmentation are continuously evolving. However, the choice between different network architectures depends on the specific requirements of the segmentation task, the available computational resources, and the size and quality of the training dataset. Overall, the results from the present study indicated that the AKU-Net model is an efficient approach for AK detection, paving the way for more effective and reliable evaluation, continuous monitoring of condition progression, and assessment of treatment responses. 

## Figures and Tables

**Figure 1 cancers-15-04861-f001:**
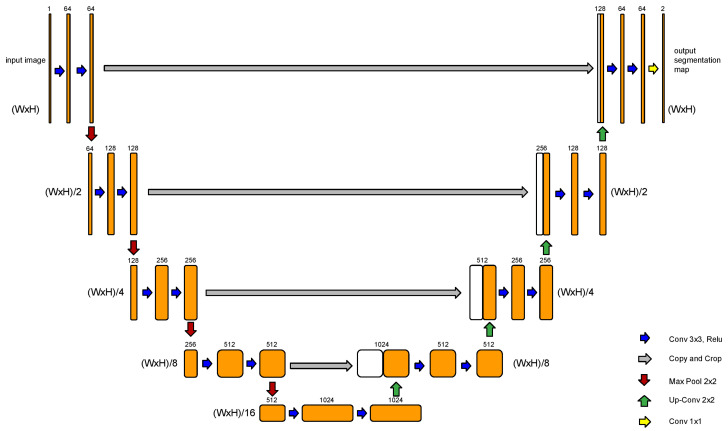
U-Net architecture proposed by Ronneberger et al. (2015) [38].

**Figure 2 cancers-15-04861-f002:**
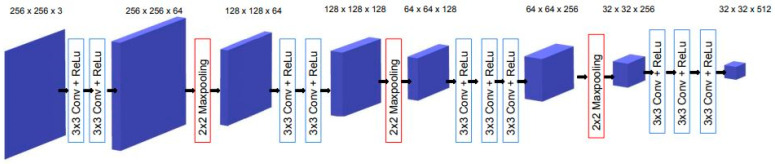
VGG16 backbone transfer learning scheme for the U-Net encoder.

**Figure 3 cancers-15-04861-f003:**
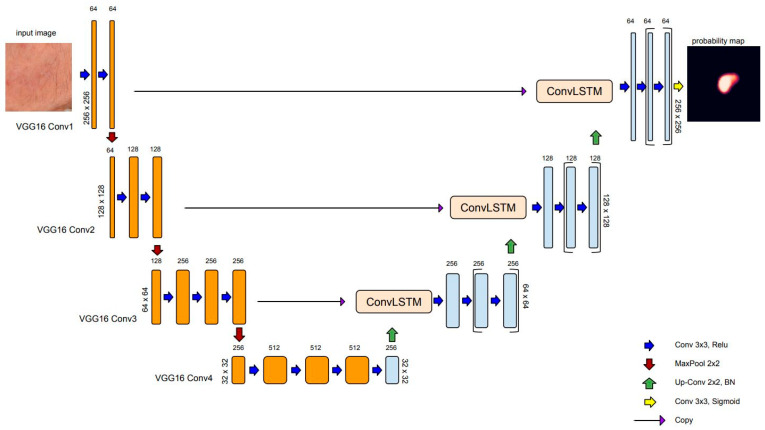
Proposed model for AK detection based on U-Net architecture (AKU-Net), with three key modifications: a utilizing the pretrained VGG16 as the encoder, incorporating convLSTM processing units in the skip connections, and integrating the BN in the decoding layer.

**Figure 4 cancers-15-04861-f004:**
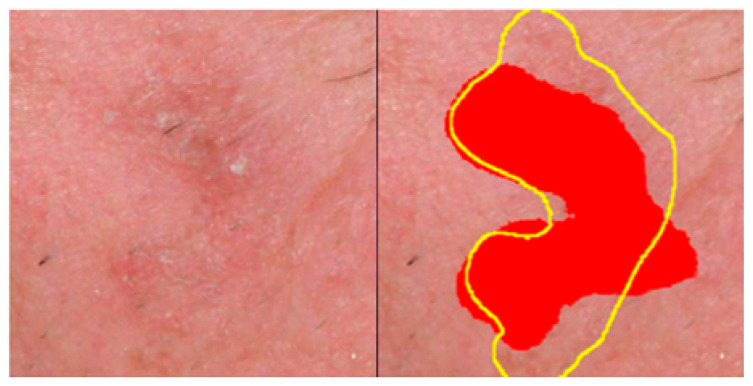
A patch from a clinical photograph with AK (**left**) and the predicted area (**right**). The AK labeling from the system is highlighted in red, and the yellow line represents the expert’s annotation. The estimations for the Dice, IOU, and aF1 coefficients were 0.76, 0.62, and 0.97, respectively.

**Figure 5 cancers-15-04861-f005:**
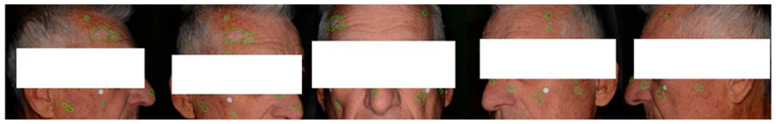
Photographs were taken to capture the presence of multiple lesions across the entire face and provide different views of the same lesions. With green color are the annotated by the experts AK lesions. The white circular sticker is a fiducial marker with a diameter of ¼ inch.

**Figure 6 cancers-15-04861-f006:**
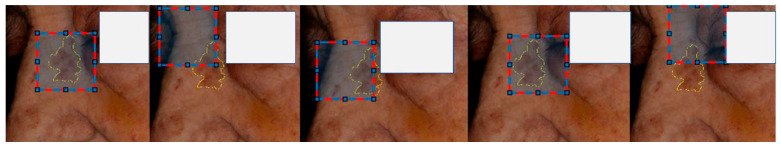
From left to right: 512 × 512 lesion center crop and the corresponding translation-augmented lesion crops.

**Figure 7 cancers-15-04861-f007:**
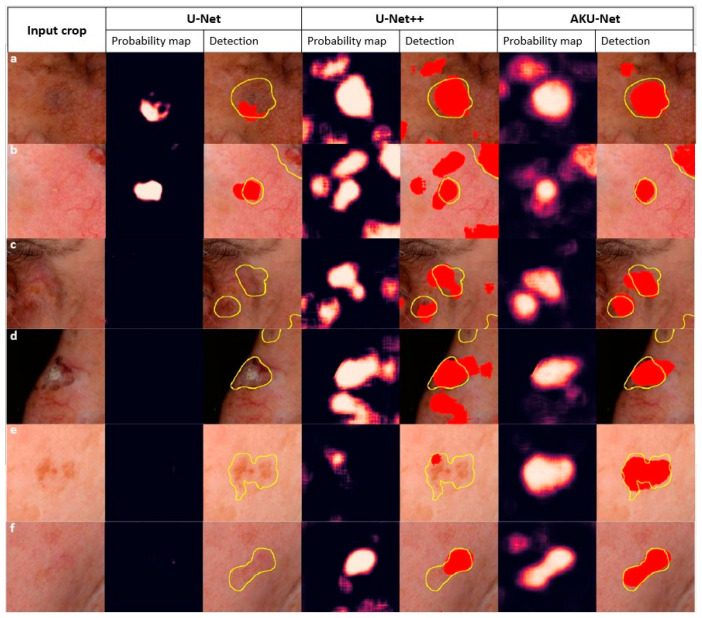
A visual demonstration of the efficiency of the three trained models in AK detection in challenging skin areas. Skin folds, hairs, and small vessels all constituted sources of severe false positives for AKCNN (**a**–**d**). AK lesions with a low contrast and ambiguous boundaries (**e**,**f**) were successfully detected by AKU-Net. Note the inclusion of lesions from almost the whole spectrum of clinical AK grades. The experts’ annotations are in yellow, and the models’ predictions in red.

**Figure 8 cancers-15-04861-f008:**
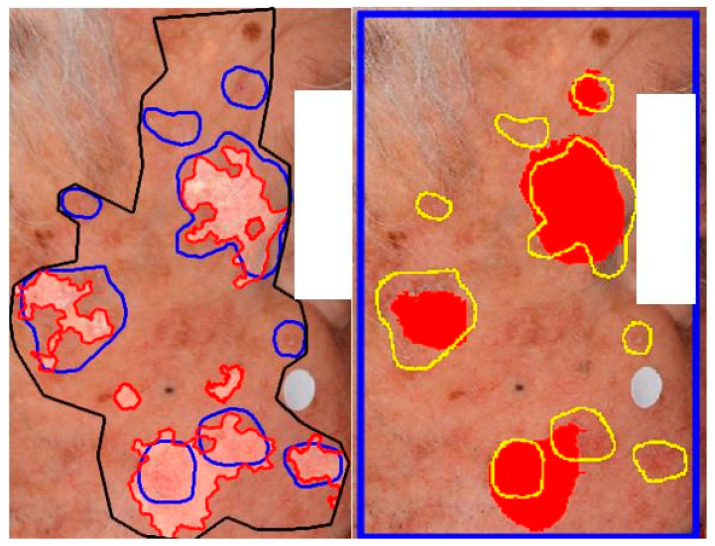
Exemplary visualization of AK detection of the same frame (Table 3; Frame 3) with two model architectures. (**Left**) The performance of AKCNN, where the “scanning” area (black line) was manually predefined to exclude areas covered by hairs and the anatomical structure of eyes: blue lines are the expert-annotated AK lesions and scanty colored areas correspond to the detected AK. (**Right**) Detection of AK using AKU-Net in the entire frame region (blue box). Note the aggregation of the AK-affected skin area in four distinct patches (red color).

**Figure 9 cancers-15-04861-f009:**
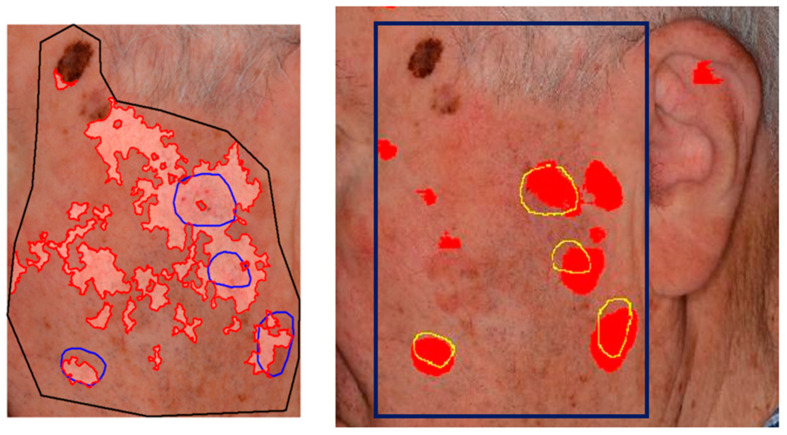
Exemplary visualization of AK detection of the same frame (Table 3; Frame 6) with two model architectures. (**Left**) AKCNN detection results with the highest false-positive rate aPrec=0.26. (**Right**) The AKU-Net was favorably tolerant of the selection of the scanning area that was simply either a boxed area (blue box; aPrec=0.71) or considered as a wider frame (aPrec=0.69).

**Table 1 cancers-15-04861-t001:** Dataset splitting into train validation and test sets.

	Patients	Images	Crops	Augmentation
Train	93	410	13,190	Yes
Validation	5	100	3298	Yes
Test	17	59	403	None
Total	115	569	16,891	

**Table 2 cancers-15-04861-t002:** Segmentation accuracy of utilized model architectures.

Architecture	Dice (Mean)	IoU (Mean)
U-Net	0.14	0.48
U-Net^++^	0.39	0.55
AKU-Net	0.50	0.63

**Table 3 cancers-15-04861-t003:** Comparison of the accuracy of AKU-Net and AKCNN (*n* = 10 random frames).

	AKCNN			AKU-Net		
Frame	aPrec	aRec	aF1	aPrec	aRec	aF1
1	0.96	0.67	0.79	0.81	0.67	0.73
2	0.77	0.6	0.67	0.56	0.50	0.53
3	0.77	0.56	0.65	0.94	0.56	0.70
4	0.5	1	0.67	0.88	0.80	0.84
5	1	0.25	0.4	1.00	0.50	0.67
6	0.26	1	0.41	0.69	1.00	0.81
7	0.7	0.67	0.68	0.33	0.67	0.44
8	0.86	1	0.93	0.73	1.00	0.84
9	0.99	0.27	0.43	0.74	0.45	0.56
10	0.96	0.6	0.74	0.60	0.60	0.60
Median	0.82	0.64	0.67	0.73	0.63	0.68

## Data Availability

The developed code is available at the following link: https://github.com/PanagiotisDerekas/Skin-lesions/tree/PanDerek/Actinic%20Keratosis (accessed on 1 October 2023).

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
