# Peer review of "The Promise of Semantic Segmentation in Detecting Actinic Keratosis Using Clinical Photography in the Wild"

_cancers, 2023, doi:10.3390/cancers15194861_

Round 1
Reviewer 1 Report
Excellent paper presenting an automated algorithm for the clinical detection of AKs.
Some minor revisions for the authors:
1. Introduction: Concerning diagnosis of AK the authors do not refer to dermoscopy which is an everyday diagnostic tool for the majority of dermatologists.
2. Material: Could you please describe the clinical staging of the AK lesions of the patients? Thin AKs (Olsen staging I) are not easily detected macroscopically. Palpation is required and dermatoscopic examination is always of help. Thus, the AK lesions included in the study were typical, thicker AKs with clinically obvious scale (Olsen stage II-III)? If yes, this could be a limitation of the system since it was trained on a subset of AKs.
3. Concerning the selection of the lesions did the authors exclude clinically ambiguous lesions? Or did they choose to follow with a biopsy? Please clarify.
4. Was the training model used only for non-pigmented or also for pigmented AKs? Pigmented AKs are a differential diagnostic pitfall for lentigo maligna and a biopsy is required.
Author Response
We would like to thank the reviewers for their supportive feedback and their valuable comments.
Reviewer1:
- Introduction: Concerning diagnosis of AK the authors do not refer to dermoscopy which is an everyday diagnostic tool for the majority of dermatologists.
Answer: We have added a paragraph that addresses this comment, along with the suggestions from the third reviewer:
Introduction, lines 61-68 :“ In the everyday clinical setting dermatologists employ a spectrum of established non-invasive diagnostic techniques to increase the sensitivity and specificity of the clinical diagnostic workout and to improve the discrimination between AK and CSCC [3],[9]. Besides the widely available and routinely applied dermoscopy to discriminate between invasive CSSC and AK [10], recent findings show that the combination of minimally invasive histologic markers, like the basal layer proliferation score, [11] with non-invasive imaging modalities, like LC-OCT, can be used to better stratify AKs according to their risk to progress to CSCC [12].”.
- Material: Could you please describe the clinical staging of the AK lesions of the patients? Thin AKs (Olsen staging I) are not easily detected macroscopically. Palpation is required and dermatoscopic examination is always of help. Thus, the AK lesions included in the study were typical, thicker AKs with clinically obvious scale (Olsen stage II-III)? If yes, this could be a limitation of the system since it was trained on a subset of AKs.
Answer: We have modified the introduction, lines 50-51, lines 69-83 and lines 135-137 (aim), including the appropriate references [3],[16],[17] , so to emphasize the clinical problem which is the AK burden quantification in skin cancerization fields, and our aim which is to support the quantitative evaluation of AK burden using clinical photography and thus to contribute to the development of reliable instruments to evaluate AK and, consequently SCF burden to be primarily used in the evaluation of therapeutic interventions. In this context, in our study we have included AK lesions with a wide spectrum of clinical characteristic as far as can be discerned by the experts by inspecting the clinical image. It is important to emphasize that the utilization of cross-polarized photography enhances significantly subtle clinical characteristics of AK lesions and increases the “sensitivity” with which an expert demarcated the AK lesions for training our model.
Moreover visual examples are provided in Figure 7 (updated caption: Note the inclusion of lesion from almost the whole spectrum of clinical AK grades)showing that the model can detect a variety of AK lesions from subtle AK, and AK with pigmented characteristics to thicker AKs with clear characteristics but with a challenge to be detected from an automatic system as they appear to be near the eye and/or hairy parts of the face or are surrounded with small vessels (vascular plexus).
Also, in the ‘Material and Methods’ section, lines 360-369, we have added following paragraph (with ref. [61]) clarifying further the eligible criteria of AK cases selection:
“Notably, cross-polarized photography was employed that enhances the visibility of the vascular plexus of the skin (redness) and removes unwanted glare from the epidermis, allowing the detailed evaluation of the assessed area in larger magnifications, as required. For the purposes of the present study, figures were eligible only from patients without history of CSSC and from areas without ambiguous lesions, including pigmented lesions, clinically susceptive for cutaneous malignancy. Assessments of the clinical grades of individual lesions are not reported, as are out of the scope of the present study. However, we incorporated lesions independent of clinical AK severity grading [61] so far the lesions could be suspected on the selected clinical photographs, as well as lesions with variable degrees of pigmentation. This also applied to AKs in extra-facial anatomical regions.”
- Concerning the selection of the lesions did the authors exclude clinically ambiguous lesions? Or did they choose to follow with a biopsy? Please clarify.
Answer: Thank you for this critical comment, which gave us the opportunity to clarify further our material selection strategy by adding following information (Material: Lines 363-365): “For the purposes of the present study, figures were eligible only from patients without history of CSSC and from areas without ambiguous lesions, including pigmented lesions, clinically susceptive for cutaneous malignancy.”
- Was the training model used only for non-pigmented or also for pigmented AKs? Pigmented AKs are a differential diagnostic pitfall for lentigo maligna and a biopsy is required.
Answer: This comment has been already answered as part of the answer in comment 2.
We believe that Figure 7 is representative to our effort to include AK lesions from a wide spectrum of clinical characteristics including the pigmented characteristics.
Reviewer 2 Report
A straightforward paper with innovative approaches not only to detect actinic keratoses but to use AI for recognition based on small sample sizes.
Author Response
We are indebted to Reviewer 2 for the encouraging comments.
Reviewer 3 Report
The authors submitted a manuscript addressing the potential of semantic segmentation of AK lesions to improve the monitoring of AK burden in clinical settings. Given the high prevalence of AKs and the rise of artificial intelligence applications within the dermatological field, the authors report on a very relevant topic.
The methodology of this study appears to be well-structured and provides clear information about the study's objective, the methods used, and the study population. Especially for continuous and objective monitoring and assessment of treatment responses of AK lesions this method shows promising results.
However, it is important to identify potential limitations:
1) line 55: recent findings show that histologic PRO score correlates with progression to SCC (see Schmitz L. et al.: „Evaluation of two histological classifications for actinic keratoses - PRO classification scored highest inter-rater reliability“). Furthermore non invasive imaging using LC-OCT facilitates in vivo PRO score assessment, resulting in the possibility for monitoring AK progression and treatment results up to cellular level (see Daxenberger F. et al.: „Innovation in Actinic Keratosis Assessment: Artificial Intelligence-Based Approach to LC-OCT PRO Score Evaluation“)
2) For the diagnosis of actinic keratoses not only macroscopic images but more important dermoscopic view and haptic feeling of hyperkeratoses is most important.
Moreover, areas difficult to access (e.g. retroaurikular or covered by hair) cannot be evaluated from the picture alone and therefore results can be false negative. Given that methodologic limitations this technique should be interpreted with caution as a possible addition for objective follow-up but will not replace a human clinical and dermoscopic examination.
Author Response
We would like to thank the reviewers for their supportive feedback and their valuable comments.
Reviewer 3
- line 55: recent findings show that histologic PRO score correlates with progression to SCC (see Schmitz L. et al.: „Evaluation of two histological classifications for actinic keratoses - PRO classification scored highest inter-rater reliability“). Furthermore non invasive imaging using LC-OCT facilitates in vivo PRO score assessment, resulting in the possibility for monitoring AK progression and treatment results up to cellular level (see Daxenberger F. et al.: „Innovation in Actinic Keratosis Assessment: Artificial Intelligence-Based Approach to LC-OCT PRO Score Evaluation“)
- For the diagnosis of actinic keratoses not only macroscopic images but more important dermoscopic view and haptic feeling of hyperkeratoses is most important.
Answer: We have modified appropriately the ‘Introduction’ including the suggested references:
Introduction, lines 61-65 :“ In the everyday clinical setting dermatologists employ a spectrum of established non-invasive diagnostic techniques to increase the sensitivity and specificity of the clinical diagnostic workout and to improve the discrimination between AK and CSCC [3],[9]. Besides the widely available and routinely applied dermoscopy to discriminate between invasive CSSC and AK [10], recent findings show that the combination of minimally invasive histologic markers, like the basal layer proliferation score, [11] with non-invasive imaging modalities, like LC-OCT, can be used to better stratify AKs according to their risk to progress to CSCC [12].”.
- Moreover, areas difficult to access (e.g. retroaurikular or covered by hair) cannot be evaluated from the picture alone and therefore results can be false negative. Given that methodologic limitations this technique should be interpreted with caution as a possible addition for objective follow-up but will not replace a human clinical and dermoscopic examination.
Answer: We would like to thank the reviewer for these comments, which has prompted us to provide further clarification in the Introduction regarding the clinical problem and the aim of the present study. The clinical problem at hand is the quantification of AK burden in skin cancerization fields, and our objective is to support the quantitative assessment of AK burden through clinical photography. This contribution aims to aid in the development of reliable tools for evaluating AK and, consequently, field cancerization burden and we hope these tools will primarily serve in the evaluation of therapeutic interventions in the future.
For this, we have appropriately modified the introduction, lines 50-51, lines 61-83 and lines 135-137, including the appropriate references [3],[16],[17], so to emphasize the clinical problem. In this context, in our study we have included AK lesions with a wide spectrum of clinical characteristic as far as can be discerned by the experts by inspecting the clinical image. At this point we would like to stress that the use of cross-polarized photography enhances significantly subtle clinical characteristics of AK lesions and increases the “sensitivity” with which an expert demarcated the AK lesions for training our model.
We believe that Figure 7 is representative of our efforts to include AK lesions with a wide spectrum of clinical characteristics, even in challenging facial areas (e.g., Figure 7; panel c)
Also, in Discussion Lines 488-499 we have added the following text that further discuss our focus on clinical photography:
“The palpation of the lesional skin to confirm the characteristic ‘sandpaper’ sign is crucial for the diagnosis of barely visible flat, grade I AKs. However, the fact that the present approach relies on photographic material might theoretically lead to underestimation of the AK burden does not represent a serious limitation of this method. It is worth noting that the ‘sandpaper’ sign has been ranked as a less reliable feature of the sun damage of SCF skin areas compared to ‘visible’ features (telangiectasia, atrophy, and pigmentation disorders) in a panel of experts’ study [3]. Moreover, the proposed approach, like AK-FAS too [15], primarily aims to quantify AK burden in selected skin areas to assist the evaluation of therapeutic interventions. Accordingly, the burden measurements are based on the evaluation of index lesions in the preselected target area. If required, the latter can be planned to include ‘hidden’ areas, like the retroauricular area or skin regions covered by hair”